# “Ultimately, You Realize You’re on Your Own”: The Impact of Prostate Cancer on Gay and Bisexual Men Couples

**DOI:** 10.3390/ijerph20105756

**Published:** 2023-05-09

**Authors:** Joseph Daniels, Rob Stephenson, Shelby Langer, Laurel Northouse, Roxana Odouli, Channa Amarasekera, Stephen Vandeneeden, Marvin Langston

**Affiliations:** 1Edson College of Nursing and Health Innovation, Arizona State University, Phoenix, AZ 85004, USA; 2University of Michigan School of Nursing, University of Michigan, Ann Arbor, MI 48109, USA; 3Division of Research, Kaiser Permanente of Northern California, Oakland, CA 94612, USA; 4Department of Urology, Feinberg School of Medicine, Northwestern University, Chicago, IL 60611, USA; 5Department of Epidemiology and Population Health, School of Medicine, Stanford University, Stanford, CA 94305, USA

**Keywords:** prostate cancer, gay men, bisexual men, relationship, couples, communication

## Abstract

An estimated one in three gay and bisexual (GB) male couples receive a prostate cancer (PCa) diagnosis over their life course with limited understanding of the impacts on their relationships. Psychological distress related to PCa diagnosis and treatment-related side effects have been shown to disrupt established GB partnership dynamics. Communication barriers often develop within GB relationships affected by PCa, further exacerbating couple tensions, isolating partners, and lowering quality of life for both patients and partners. In order to elaborate on these phenomena following a PCa diagnosis, we conducted focus group discussions with GB men in relationships. Men were recruited nationally through PCa support groups, and after completing consent procedures, they were invited to one of two focus group discussions conducted through video conference. Topics discussed included the diagnosis and medical decision making pertaining to PCa; healthcare provider experiences; the emotional, physical, and sexual impact of PCa diagnosis and treatment; sources of support and appraisal of resources; and partner involvement and communication. There were twelve GB men who participated in focus group discussions that were audio-recorded and transcribed, and analyzed using a thematic approach. GB couple experiences with PCa during and after treatment choice and recovery identified common patient–provider communication barriers. In particular, GB men reported difficulties in disclosing their sexuality and relationship to their providers, limiting conversations about treatment choice and partner engagement in care. Both patients and partners experienced times of being alone after treatment, either by choice or to give space to their partner. However, partners often did not explicitly discuss their preferences for being alone or together, which resulted in partners’ disengagement in their relationship and the prostate cancer healthcare process. This disengagement could blunt the notable PCa survival benefits of partnership for GB men.

## 1. Introduction

In the U.S., an estimated one in six gay and bisexual (GB) men and one in three GB same-sex couples receive a prostate cancer (PCa) diagnosis during their lives, which can negatively impact their relationships [1,2,3,4,5,6,7,8]. Numerous and intersecting biological and psychosocial effects can occur after PCa post-treatment, including urinary and sexual dysfunction that can alter partnership dynamics [1,4,6]. Specifically, PCa treatment side effects have been shown to reduce intimacy between male couples, with unique challenges that center on GB sexual roles—insertive and receptive roles in anal intercourse [9]. When it comes to sex, GB men with PCa report higher rates of erectile dysfunction, rectal discomfort, and loss of sensitivity when compared to heterosexual men, which can disrupt their sex lives in a relationship [4,10]. In two studies to assess PCa experience among GB men, 22% [11] and 27% [9] reported difficulty having erections to conduct anal intercourse, and 33% reported anal pain during sex [11]. To cope, GB male couples have experimented with switching anal sex roles [12]. However, such physical and role changes have a profound impact on identity in that GB men report PCa diagnosis increased gay identity stress due to anticipated or experienced treatment side effects [8,13,14]. In addition, unanswered questions remain regarding GB male-specific recommendations on timelines to safely return to anal sex following treatment given the possibility of rectal inflammation, bleeding, or residual radiation from brachytherapy. Post-treatment sequelae such as erectile disfunction and rectal discomfort are difficult to discuss among GB patients and partners [5,15,16,17]. As a result, GB men report lower quality of life, lower self-esteem, and greater psychological distress (i.e., internal and external sexuality-related stigma); with their partners reporting lowered quality of life as well [2,5,18]. 

From the broader literature on couple functioning within the context of cancer, research indicates that supportive patient–partner (i.e., dyadic) coping helps them to navigate shifts in their sexuality and intimacy, and is associated with better outcomes for both patients and partners [13,19,20,21]. For example, partners’ sexual interest is positively associated with patients’ recovery of sexual function [22]. Similarly, partners’ level of depression is predictive of the patient’s relationship satisfaction, sexual satisfaction, and perceived quality of communication [23]. Despite the dyadic health benefits that can be leveraged from the inherent interdependence in couples’ relationships, GB couples are not benefitting to the extent possible [22]. 

In part, these individual and relationship outcomes may stem from barriers in patient–provider communication. Specifically, many GB same-sex couples managing chronic conditions or cancer are often reluctant to go together to clinics due to anticipated or enacted stigma and discrimination in clinics that may compromise treatment [24,25,26]. As a result, GB couples often do not receive the same education and clinical information at the same time [27], limiting communication and support that in turn leads to isolation [28]. For PCa, this is substantiated by a survey of 112 academic urologists that found most clinicians reported not asking patients about sexual orientation, and over a quarter presuming all patients to be heterosexual at first encounter [29]. Clinicians also felt significantly more comfortable discussing sex with heterosexual patients and had significant knowledge gaps on the unique health needs of sexual minorities [29]. These patient–provider outcomes may be attributed in part to limited GB health competency training provided to medical specializations, such as urology or oncology [26,30], and also to limited patient education about the importance of partner engagement in treatment combined with limited examples of how to disclose GB identity to providers to facilitate partner engagement. These barriers hinder the ability of patients, partners, and providers to bridge communication gaps to explore treatment choices together [30,31,32]. Such experiences can isolate partners from one another, resulting in communication barriers that further exacerbate tension within couples, and in turn lower quality of life for both partners [3,24].

Although there is a growing body of knowledge that outlines the barriers and facilitators for GB men’s PCa treatment outcomes [4,7], more research is needed to understand the PCa diagnosis and treatment experiences of GB patients and their male partners. Therefore, we conducted a qualitative study to explore GB patients’ and partners’ perspectives and experiences on relationship dynamics and clinical engagement during PCa diagnosis, treatment, and recovery. 

## 2. Methods

### 2.1. Study Design and Participants

Two focus group discussions were conducted among GB men with PCa and their same-sex partners. Our nationwide recruitment strategy was a two-step process. First, administrative staff of prostate cancer support groups for GB men were contacted via email and asked to share an advertisement with their members about a study designed to explore the experiences of GB men with prostate cancer and their same-sex partners. Included in the advertisement was that study results could be used to help design a couple-based intervention to provide GB-tailored PCa information and support. Respondents to this advertisement then reached out to the study team via email to confirm interest. Next, a videoconference call was setup with the study team to determine eligibility. Participants had to be diagnosed with PCa and identify as gay or bisexual, or be a romantic partner of a man with PCa diagnosis. If eligible, men were invited to complete the informed consent process, and then scheduled for one of two focus group sessions. Focus groups were conducted on MicrosoftTeams^TM^ to allow for nationwide inclusion. Focus groups included a total of 12 GB men.

### 2.2. Data Collection

Focus groups were scheduled for 1.5 h, and cofacilitated by two researchers (one with expertise in cancer epidemiology, M.L., and the other with expertise in health outcomes interventions among GB men, J.D.) using a semistructured protocol, allowing for participant-driven inquiry. Major domains of exploration included the following: the diagnosis and medical decision making pertaining to PCa; healthcare provider experiences; the emotional, physical, and sexual impact of PCa diagnosis and treatment; sources of support and appraisal of resources; and partner involvement and communication. Participants received USD 25 Amazon gift cards. Focus groups were conducted using Microsoft Teams videoconferencing software and recorded (voice only) for transcription. Transcription was initially completed by the Microsoft Teams software and manually corrected by study team staff. 

### 2.3. Data Analysis

Using thematic analysis, transcripts were reviewed and then a subset were coded to identify GB couple and patient–provider experiences during PCa treatment and recovery [33,34,35]. After review by the research team, codes were assembled into a codebook that was applied to all transcripts. Once coding was completed, a frequency analysis was conducted to generate a range of GB couple experiences from most to less frequently discussed, which were subsequently organized into matrices to understand positive and negative influences on PCa treatment [27,36]. Memo writing and causal diagrams were developed to refine preliminary themes that were presented to the research team for discussion, informing additional analysis to generate the final themes [37,38]. 

### 2.4. Ethics and Participant Representation

Study procedures were approved by the Institutional Review Boards of Kaiser Permanente Northern California (1793016-4) and Arizona State University (00014559). In order to protect confidentiality, all participants with PCa are represented by treatment choice (radical prostatectomy, RP, or active surveillance, AS) and numbered consecutively. All partners are represented, P, and numbered with partner treatment choice. 

## 3. Results

There were twelve (N = 12) GB men who participated in the focus group discussions. There were two couples, seven men whose partners did not participate, and one man who was not in a relationship at the time of data collection. Ten of the participants attended PCa support groups for GB men. Participants were from Toronto, Los Angeles, San Francisco, San Diego, Palm Springs, and Washington DC. The age range for the participants was 55–74 years with initial prostate cancer treatment or diagnosis (for the one man on active surveillance) occurring between 6 months to 10 years prior to eligibility session. Major themes show that patient–provider communication regarding sexuality, treatment regret, and miscommunication in the relationship shape the PCa experience for GB couples. 

**Sexuality and identity increasingly become important in provider communication for treatment choice and partner engagement in care.** Choosing a urologist for GB men and their partners was an involved process of consulting with other gay men and those in their social networks for advice. Some participants remarked that the sexual orientation of the provider was important, yet others stated this was unrealistic since there are few gay urologists in their area. However, many participants described needs for providers who can deliver affirming care given their sexual orientation or appear sympathetic. 


*“A lot of guys in my support system and support group will not go to a straight doctor period. They have to know that the doctor’s gay before he even goes to him. In my case, I’ve got a urologist who is uh very gay positive but not gay. So, you know, it’s not a problem for me. So, it just depends on your relationship [with your] doctor.”—RP5*



*“There was never a question of trying to find either a gay urologist or gay-friendly urologists. [It] just wasn’t going to happen in the community. So, our task [as a couple] was to find the best urologist for the job.”—RP7*


Nearly all GB men discussed wanting to disclose their sexual orientation with their chosen provider. However, this disclosure was beneficial for some but not all. Overall, participants stated that even after disclosure they continued to feel that their providers were not able to engage with or tailor discussions to how treatment might uniquely impact GB men, including the effects on their sex life and sexual identity. 


*“To me [disclosing sexual orientation], that’s just a part of my health history. Yeah, I’m 5 foot 8. I weigh 175. Uh, and I’m gay. This is just consistent with most things, making sure that the provider was going to be gay friendly.”—RP6*



*“I walk in there with my partner, or I walked in alone, and I mentioned my partner…You know it’s all very open and almost no judgment. My urologist is very gay friendly.”—RP5*



*“I guess the only issue with the urologist- I think he was uncomfortable talking about the mechanics of gay sex. So, when you ask questions about things [how different treatments impact sex life], he was a little uncomfortable. But, he tried his best [with us], and he did refer us to a sexual function therapist.”—RP7*



*“I certainly would have preferred to have a really honest conversation with a practitioner who was gay or bisexual, who had some awareness of what it’s like to have sex with men, and some of the, you know, it’s differences…”—RP1*


Participants with PCa and their partners discussed the importance of disclosing their sexuality to their provider, not only to ensure competent care but also to open the door to frank conversations about treatment choice that may impact their sexuality and identity.

**Treatment uncertainty and decisional regret compounds difficult relationships with providers and in turn impacts relationships.** After choosing an initial provider, GB men and their partners undergo an intense period of treatment uncertainty. This is attributed to experiences with providers who are described as coercive, emphasizing a particular treatment, reducing choice, and seemingly not having the patient’s best interest at heart. As result, many GB men and their partners described obtaining second and third or more opinions.


*“Uh, I was diagnosed 18 months ago. I’ve had seven different opinions about what I should do. So far I haven’t done anything because the PSA went from 1618 down to 7 and the doctors, at least my primary care says, ‘Just wait and see.’ I’ve read everything I could get my hands on… And I sit here having done nothing, and now not quite knowing what to do. I am seeing another urologist in [city] in a couple of weeks. He was the one that told me not to do anything initially….All the tests have been done showing no spread. And as a man in his late 70s, I am constantly concerned there are some symptoms, but there’s no pain. The symptoms are…not a strong urine stream, and the fact that I can’t pee and poop at the same time. But, there again there’s no pain, and there is some sexual functioning. So I’m in a place where I don’t know really what to do. I do know Doctor [name], and he is a source of support, but otherwise I’m pretty much alone.”—AS1*



*“That every specialist is an advocate for that [particular] treatment. So, you don’t get an unbiased opinion anyway from anyone. I saw three different [urologists]. Uh, I sought two different second opinions. One in Baltimore, and two in Washington DC, and they’re all just advocates for their procedures. I felt, we felt like they were just advocates for their process as opposed to saying this is best for you because of these sets of criteria.”—RP5*


Ultimately, many GB men identified a need to educate themselves more, and in many ways, become their own best advocates. Given the limited available information about treatments and side effects that may impact sexuality, their identity, and their relationship, many GB men discussed regretting their treatment decision. 


*“*
*The other thing, I think you need to ask and be your best own advocate. Because [it may be that] the morning after the surgery, the doctor shows up with his ski gear on, and he says, ‘You’re just fine. Everything is just perfect. You know, we’ve saved all the nerve endings. You’ll not be incontinent.’ It was all a total lie.”—RP2*



*“I mean, you might have somebody who says, ‘Oh, I came out of this radiation. I’m fine.’ And a year later, the problems start. Uh, so that you should also keep that in mind that it’s really not the cancer that’s the problem. It’s the modality of treatment that you’ve chosen.”—RP5*



*“I’ve got these seven choices. I’ve gotta pick one ‘cause I want to get this cancer out of my body...Nobody [urologist] was talking about active surveillance as really a much of a viable option. I mean. Or you could wait…as I look back now, and also in talking to other people, I didn’t really realize the huge impact it’s going to have on my life, and that active surveillance might have been a very reasonable choice instead of the robotic.”—RP8*


Some patients wish their sexuality and identity would have been included in the treatment decision so as to prevent any sexual functioning and urinary side effects, or to establish a mechanism to discuss and treat in post-treatment, if side effects develop. 


*“Then, as opposed to my urologist, who is straight and couldn’t really identify with you, learning from them about some of the results of the surgery might be from a [GB] experience perspective. What that might mean for me, you know if you’re a top or a bottom? You know, what will happen, right?”—RP1*



*“I know that for my husband, his first three months were focused on his scar. And, I know that sounds silly, but he was just so convinced that it all going to fall apart. So to try and have him [to] even think about sex [during that time] would have been impossible. Uh, now, and a year later, it’s on his mind. But, there’s really nobody to talk to about it.”—P7 (Partner RP)*


As a result, many participants in the study discussed creating or joining post-treatment support groups that focused on the GB experience.


*“So, in an effort to be more frank about our GB experiences [after treatment], we decided to, you know, to start a support group. It’s been a very frank and very open place for honest discussion. We’ve had, you know, show and tell of peoples, you know, what kind of underwear they’ve worn or incontinence, and what type of [penis] pump they’ve used, or what type of devices they’ve used for any stimulation that they can. You know they find it is beneficial to them.”—RP4*


**Miscommunication and a lack of partner PCa education led to “going it alone” at times.** Both men with PCa and their partners described differing roles and responsibilities regarding diagnosis and treatment. Many described having to educate their partners about prostate cancer, which became exhausting and exacerbated by side effects that limited couple-based intimacy and discussions about intimacy with partners. 


*“So. It’s been a real, long struggle teaching my partner [about prostate cancer] and finding other support systems where we are in our group to share our experiences and watch training films [on prostate cancer].”—RP2*



*“It’s also about performance [sex]. That soon after treatment, you’re still probably struggling with things like having to pee. I peed in the bed, I mean, or erectile dysfunction. What do you mean by sex? I’m trying to just get a good snug goodnight, snuggling, and then getting up and changing the sheets because you peed in your pants. It’s not a full-on sex. We’re [couple] not talking about a whole lot of that going on, right after the, uh, the surgery.”—RP5*


Despite this education, GB men with PCa discussed partner support as less than ideal at times, yet some partners made it up in other ways, explaining that at times they were responding to their partner’s needs to have space for a while. 


*“*
*The support was [that my partner was] not engaged in the decision-making as much as compared to other people. But also, you know, I never saw a bill. I mean the insurance companies sent, you know, did their whatever. He just took care of all of that.”—RP5*



*“Yeah, he needed time away. Needed time on his own. And it’s my job to give it to him, I guess. So, no, it wasn’t an issue. But you know, I’ve not had this, so it’s hard. It was hard for me to know exactly what he was going through. So, I took my cues from him.”—P7 (Partner RP)*


Some participants described that ultimately their treatment choice was their decision alone, though they said on the flip side that “*couples survive the surgery”*. Treatment recovery was compounded by other chronic diseases that couples were managing, such that participants stated that they made choices about the degree to which they involved their partner based on the partner’s health issues and perceived stress. 


*“We [couple] were in a little bit different situation. My husband was in renal failure simultaneous to my surgery. We’re looking at the placement of the port for dialysis, and the timeline was shifting quickly on his renal care. Um and so, and needless to say, that was not emotionally going well for him or me. So, for me to talk about this with him, just it would add something that just I was concerned that would not go well. And as it turned out, there was, there was a lot more renal trauma later. Uh, and still continues to be so. I involve him very little in my whole treatment. I’m having an artificial sphincter put in for a year and a half of bladder failure. So, it’s just. It is what it is.”—RP6*



*“Ultimately, you realize you’re on your own. You’ve got to make that decision yourself [treatment choice].”—RP5*


## 4. Discussion

Our study provides perspectives on how GB men with PCa and their partners navigated care and post-treatment effects. While study participants noted disruptions in intimacy due to the PCa care experience, different coping strategies were reported, such as self-isolating or taking time to be alone without communicating with their same-sex romantic partners. Although patient–provider communication was impacted often due to medical mistrust, GB men desired to engage with their provider in discussions about their sexual identity and relationships when making decisions about treatment choice. 

Nearly all participants discussed long-term or lifetime side effects post-treatment, such as incontinence and erectile dysfunction. As a result, there was treatment choice regret among them, with at least one participant losing trust in his urologist because of permanent damage that occurred, during which time the urologist demonstrated limited empathy. While treatment regret is common for all men irrespective of sexual orientation [39], many GB men in our study strongly felt that these experiences may be ameliorated if they could talk with a urologist who was gay and had the perspective that gay sex and identity are linked. According to study participants, they could have made more informed treatment choices if they were able to talk openly with the urologist about their concerns and needs post-treatment [14,40]. These types of conversations would help GB men prepare for upcoming negotiations around sexual role switching and changes in sexual identity. However, as mentioned previously, no GB men described this opportunity to work with a gay urologist. 

Few partners were described as fully engaged in PCa treatment and recovery, though support was given in the form of taking care of bills or managing other aspects of their lives as a couple. Partners were often described as needing more PCa education to provide support as well, which may be in part due to the fact that they were often not attending clinic visits, as a most reported difficulties discussing GB-related health perspectives with urologists. Further, intimacy was severely affected, and most GB men discussed the difficulties of conversations around intimacy with each other. This disruption in intimacy led to changes in the sexual relationship as well. Ultimately, this shift in relationship closeness was compounded by miscommunication, increasing both partners’ isolation in dealing with PCa. Both PCa patients and partners surmised that each person needed space, with few instances of them discussing the need for this space. GB men described times of depersonalization and disengagement in the relationship as a result. In part, this relationship state has its origins in the PCa care system that has limited GB health competency, limiting couple comfort in PCa engagement from the beginning [14,27]. Further disengagement may possibly blunt the notable survival benefits of partnership following PCa diagnosis, which has been reported for married heterosexual men [41]. This disengagement is further compounded by studies demonstrating increased social isolation as well [40]. The GB community serves as a significant resource, yet GB men with PCa report feeling isolated from the GB community [40], limiting community support as they navigate treatment and post-treatment. 

As in other studies, experiences of medical mistrust were present among most GB men and their partners as few described an openness to discuss their sexuality and relationships with their urologist [6,27]. Although some men discussed having GB-friendly urologists during their PCa treatment, which increased their comfort level, others discussed having to insert their sexuality into the PCa treatment conversation themselves, irrespective of physician comfort with LGBTQ health issues and their knowledge thereof. Hesitancy to disclose their sexuality to their urologist was also noted unless the urologist was gay, but no participant described having a gay urologist. Moreover, across all participants, none of the GB men described their urologist initiating conversations about sexuality and identity as part of the treatment plan. Specifically, GB men want to understand the potential of post-treatment sexual dysfunction and sexual identity, so as to understand its influence on their sexual role in their relationship and intimacy with their partner [5,6,7]. For example, concerns around their identity as the insertive or receptive partner during sex may impact the initial treatment decision when conducting an appraisal of the potential side effects (erectile dysfunction, incontinence, bowel dysfunction, etc.) most prevalent with each treatment decision [13].

GB men find it difficult to disclose their sexual orientation to urologists [14], which is linked to the history of HIV and sexuality stigma and discrimination experience [1], and most GB men and their partners in this sample ultimately did not feel fully comfortable doing so either. Providers may not ask about sexual orientation, as 78% of clinicians in a nationwide study felt that patients would refuse disclosure and 80% felt that patients would be offended by the question [42]. However, the reason to not come out and discuss partners is less about patient comfort; GB men will not come out to health professionals to ensure that they receive the best [14,27], most informed treatment since they want to guard against discrimination, even if it is anticipated and not ever experienced. Such decisions leave partners at the periphery of care or out of the treatment decision-making process entirely at times. In our study, it was unclear if couples made a strategic decision to not attend clinic visits together because of anticipated stigma or discrimination that might impact treatment, but many GB men with PCa discussed going to these visits alone. Further, in such cases, they discussed the need for a trusted advocate that was not their partner, who could broker between the urologist and themselves for treatment decision making, due to limited GB couple–provider communication. However, treatment decision making can be a multifactorial decision with vast uncertainty and regret for sexual majority men as well [43].

## 5. Implications

More research is needed to understand GB couple engagement in PCa treatment and recovery and how this influences individual and relational wellbeing. Our research showed that partner disengagement in PCa clinic visits is possible and has detrimental effects on quality of life for both in the relationship. Specifically, if partners of GB men with PCa do not gather the PCa education and experiential learning at the same time through clinic visits and conversations with providers, limited relationship communication about care and support needs may arise, leading couples to be disengaged. This often results in periods of partner isolation and depersonalization. Prospective, longitudinal examinations of GB patient and partner experiences over time, from diagnosis through survivorship, may provide insight into relationship dynamics influencing PCa care and recovery. 

Further, given the impact of PCa on couple intimacy and communication, new or tailored interventions are warranted to engage and support both partners in treatment and recovery. Prostate cancer care systems can benefit from such interventions to improve GB same-sex couple engagement, which may in turn harness the benefits of a relationship in supporting treatment recovery and quality of life [21]. Specifically, we found that GB couples valued healthcare providers who were GB and/or familiar/sensitive to the unique challenges of GB men with PCa. Since limited data and resources on the GB PCa care experience are available, few providers in the US are able to provide such tailored care. It remains critical to educate providers, patients, and caregivers and equip them with the tools and resources necessary to provide this care [32]. Exemplars of this approach include institutions such as Northwestern Medicine who have developed urology care programs tailored for GB men [44]. In addition, research should assess the impact of previously implemented competency trainings to establish models of cancer care for gender and sexual minorities in general. Finally, our study revealed that GB men demonstrated modes of resiliency such as establishing support groups that may be assessed for implementation in clinical settings, or determining effective referral of GB men and their partners for support and reducing the isolating effect of PCa on a couples’ life.

## 6. Limitations

While our study has several strengths, such as the use of a wide recruitment and video conference delivery of focus groups that allowed for responses from men within multiple regions in the US and Canada, our results must be interpreted with the following considerations. Due to our recruitment strategy, all men were members of various GB PCa support groups. Membership in these groups may reflect men that are highly educated, more resourced, and more involved and able to navigate complicated healthcare settings than other GB men. Our study sample did not include Black men, who have higher rates of poor PCa outcomes including mortality. Future studies should evaluate the experiences of men with PCa at the intersections of race/ethnicity and sexual orientation [45]. None of the participants in our study had advanced-stage disease. The perspectives of GB men with advanced-stage disease may be distinct and this remains an underresearched area. Finally, our study sample was small for this hypothesis-generating work and likely helps support the rationale for large-scale prospective and quantitative studies that are powered to elucidate differences in the PCa experience for GB men.

## 7. Conclusions

A PCa diagnosis and resultant post-treatment side effects can negatively impact relationship dynamics for GB couples. Partner clinic visit attendance may reduce self-isolation behaviors and improve communication and support as a couple. Resources are needed to facilitate sexuality disclosure to providers which may in turn facilitate couple engagement in care, with more research needed in this area. In addition, clinic referral to community-based resources, such as GB-focused PCa support groups, may be beneficial to GB couples and patient–provider communication.

## Data Availability

The data presented in this study are available on request from the corresponding authors. The data are not publicly available due to ethics restrictions.

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
