# Peer review of "“Ultimately, You Realize You’re on Your Own”: The Impact of Prostate Cancer on Gay and Bisexual Men Couples"

_ijerph, 2023, doi:10.3390/ijerph20105756_

Round 1

Reviewer 1 Report

Well written descriptive study about concerns that GB men and partners dealing with prostate cancer have with current urology care

Author Response

Dear Reviewer: 

Thank you for your time in reviewing our manuscript. We responded to your comment below.

Well written descriptive study about concerns that GB men and partners dealing with prostate cancer have with current urology care.

Thank you very much for your support review. 

Reviewer 2 Report

Joseph Daniels etal described the work for the manuscript titled ‘Ultimately, You Realize You're on Your Own’: Ongoing Gaps 2 in Prostate Cancer Treatment and Support for Gay and Bisexual 3 Men and Their Partners Lead to Care and Relationship Disengagement' for potential publication in IJEHPH, the authors have major concerns to address before considering for publication:

1.    Authors mentioned in lines 37-39 that a correlation is associated between GB men and couples and incidence of prostate cancer, however, they did not mention any reasons or causes for such correlation from previous studies to confirm the hypothesis.

2.    Authors mentioned in line 37-39 the statistics prevalence between PCa and gay sexuality, however, this statistics not correlated to specific country, globally and to what ethnicity and racing.

3.    Authors mentioned in line 37-39 the statistics based on two types of sexuality'' GB and couples", however, they did not mention the difference in characterization and specification. And this was also mentioned in methodology without putting any selection criteria (line 93).

4.    Authors in introduction did not mention specifically why prostate cancer associated with gay sexuality has the specific side effects, although many other cancer types or other diseases may have the same impact, please specific some logic and specific reasons for the loss of sexuality and intimacy,.....etc.

5.    Writing study design has been extensively gone to details not essential to be mentioned on the expense of some important information such as number of patients, criteria of selection, classification of patients, any medications being taken that affect the selection, economic state, psychological status, ..... etc.

6.    Number of studied patient are small sample and either to increase number or clearly to mention in discussion clearly this defect of study and large scale studies are need to illustrate the effect.

7.    The authors mentioned that they keep the confidentiality of the data collected from patients, however, the sentences mentioned spoken on behalf of patients throughout the manuscript are breach this confidentiality unless they agree to publish their talks (at this case please submit some consent forms for internal reviewing showing the form of consent and patient agreement).

8.    No information about the current therapeutic options given to these patient and impact of this factor on selection and their state.

9.    Lines 126-128 did not mention the criteria differentiating between these patients.

10.The authors mention in their manuscript that the only problem for their defect health care support was to find '' gay urologist'' to understand their needs, however, this hypothesis is weak to relay on treating them in health system. Other real factors such as psychological un-balance in this age (55-70s), not finding the correct doctor (may be straight) but understand how medically and psychologically treat their cases , their religious state, family support, ..... etc. are much important to mention in the manuscript. Especially prostate cancer can affect the straight patients as well and affect their sexual life also. The patients mislead the authors, and the presence of straight doctor can solve the issue in manuscript.

11.Authors did not suggest any solutions for the current issue in health care or mention models from other different countries to compare the different health care system.

12. English editing are is required with some grammatical errors in different occasions as Line 50, Line 56, long sentences such as Lines 65-69

13. Title of the manuscript need major correction such as suggestion of '' Impact of health care system in treatment and support of prostate cancer in Gay and Bisexual men''  

14. Authors are not mentioning the full affiliation address and details.            

Reviewer 3 Report

Dear authors,

The topic is relevant and well received. It is unique and could be a good source of information for the gerala reader and health care provideers. However, there are some gaps that need to be cleared and tighten up.

1. Title - the title is not congruent with the content of the study. I find it biased as it attributes to the health care provider (Physician) the failure  in the GB couples coping towards the diagnosis/treatment and its impact to their relationship. The methods and results did not support this claim.

2.  Is there a  significant impact of PCA and its treatment unique to the GB couples and which can not be found from heterosexual couples? This has not been discussed well in the introduction or in any other part of the paper. Otherwise, readers and even the healthcare providers would not be able to relate if not verbalized or communicated well.

3. Healtcare providers respect privacy of their clients and would not intrude into their personal affairs unless the patient including the partner will be open and disclose their concerns/fears/needs. The patient can not assume that their health care provider is insensitive or is unable to meet their needs when not verbalized.

4. The participants- in order to support your title and aims, it is imperative to include health care providers among your participants. Afterall, the aim is to be able to identify and develop a healthcare support for pca diagnosed patients and their partners.

The participants involved in the FGD do not well represent the pca diagnosed patient and the partners. The inclusion criteria and sample size should be well established to ensure representativeness and generalizability of results.

5. Data analysis - the thematic analysis used should be well explained and referenced

5. Results

The results did not reveal any biases experienced for being  GB. 

Disclosure of sexual orientation was shown to be personal choice and therefore assumption that healthcare providers do not care about GB couples' coping difficulties and much more blame them for failing relationships.

Healthcare practitioners usually refer patients and their partners to experts who can facilitatite positive coping. This an only be done when the patient shows the need or expresses the need.

What goes on between the couples after diagnosis and treatment are not available to the healthcare providers unless disclosed and recognize the need for help.

The themes developed revealed the challenges experienced by the patient and their partners. Hence, the title should be worked around these themes in relation.

Overall- the methodology, results, and discussions did not meet the objectives of the study. You still can work on the current paper but focusing on the challenges experienced by the patient and the partner without reference or assumptions to the healthcare providers.

I suggest that you rewrite the article focusing on the challenges of the GB couples or include an FGD with the healthcare providers to have a complete picture of the scenario.

Thank you and good luck.

Round 2

Reviewer 2 Report

Accepted

Author Response

Dear Reviewer,

Thank you for reading and reviewing our revised manuscript. We reviewed the manuscript to complete minor English language changes and believe that these have strengthened the manuscript. 

Thank you again.

Sincerely,

Joseph

Reviewer 3 Report

Dear Authors,

Thank you for the revised manuscript. I would like you to work on a couple of things:

1. Conclusion- The conclusion should address the aim of the study based on the synthesis of your findings, furthermore add what new or significant contributions your study adds. The current conclusion can be part of the discussion or recommendation.

2. Review the citation format required by the journal.

Thank you and good luck.

Author Response

Dear Reviewer, 

Thank you for taking the time to read and review our revised manuscript. We corrected the citation style. Also, thank you for your comments regarding the conclusion. We integrated the previous version of the conclusion into the Implications section as recommended, and wrote a new conclusion responding to the criteria below. We believe the manuscript is significantly stronger as a result of these revisions.

Thank you for your time and review.

Joseph